# Clinical Validation of Cardiac Arrest Hospital Prognosis (CAHP) Score and MIRACLE2 Score to Predict Neurologic Outcomes after Out-of-Hospital Cardiac Arrest

**DOI:** 10.3390/healthcare10030578

**Published:** 2022-03-20

**Authors:** Jun-Zuo Wu, Wei-Che Chiu, Wei-Ting Wu, I-Min Chiu, Kuo-Chen Huang, Chih-Wei Hung, Fu-Jen Cheng

**Affiliations:** 1Department of Emergency Medicine, Kaohsiung Chang Gung Memorial Hospital, College of Medicine, Chang Gung University, Kaohsiung City 833, Taiwan; rebacktogoal@hotmail.com (J.-Z.W.); wei771216@gmail.com (W.-T.W.); outofray@hotmail.com (I.-M.C.); bluescratch7@gmail.com (K.-C.H.); 2Department of Emergency Medicine, Far Eastern Memorial Hospital, New Taipei City 220, Taiwan; femh91209@mail.femh.org.tw

**Keywords:** out-of-hospital cardiac arrest, neurologic outcomes, MIRACLE2, CAHP, prognosis

## Abstract

Background. Out-of-hospital cardiac arrest (OHCA) remains a challenge for emergency physicians, given the poor prognosis. In 2020, MIRACLE2, a new and easier to apply score, was established to predict the neurological outcome of OHCA. Objective. The aim of this study is to compare the discrimination of MIRACLE2 score with cardiac arrest hospital prognosis (CAHP) score for OHCA neurologic outcomes. Methods. This retrospective cohort study was conducted between January 2015 and December 2019. Adult patients (>17 years) with cardiac arrest who were brought to the hospital by an emergency medical service crew were included. Deaths due to trauma, burn, drowning, resuscitation not initiated due to pre-ordered “do not resuscitate” orders, and patients who did not achieve return of spontaneous circulation were excluded. Receiver operating characteristic curve analysis with Youden Index was performed to calculate optimal cut-off values for both scores. Results. Overall, 200 adult OHCA cases were analyzed. The threshold of the MIRACLE2 score for favorable neurologic outcomes was 5.5, with an area under the curve (AUC) value of 0.70 (0.61–0.80, *p* < 0.001); the threshold of the CAHP score was 223.4, with an AUC of 0.77 (0.68–0.86, *p* < 0.001). On setting the MIRACLE2 score cut-off value, we documented 64.7% sensitivity (95% confidence interval [CI], 56.9–71.9%), 66.7.0% specificity (95% CI, 48.2–82.0%), 90.8% positive predictive value (PPV; 95% CI, 85.6–94.2%), and 27.2% negative predictive value (NPV; 95% CI, 21.4–33.9%). On establishing a CAHP cut-off value, we observed 68.2% sensitivity (95% CI, 60.2–75.5%), 80.6% specificity (95% CI, 62.5–92.6%), 94.6% PPV (95% CI, 88.6%–98.0%), and 33.8% NPV (95% CI, 23.2–45.7%) for unfavorable neurologic outcomes. Conclusions. The CAHP score demonstrated better discrimination than the MIRACLE2 score, affording superior sensitivity, specificity, PPV, and NPV; however, the CAHP score remains relatively difficult to apply. Further studies are warranted to establish scores with better discrimination and ease of application.

## 1. Introduction

Out-of-hospital cardiac arrest (OHCA) remains a challenge, given the poor prognosis of OHCA. According to previous studies, only 2–11% of patients survive until hospital discharge in the Asia-Pacific area [1,2,3]. Several prehospital factors can influence OHCA outcomes, such as bystander cardiopulmonary resuscitation (CPR), automated external defibrillator (AED) use, the presence of a witness, emergency medical service (EMS) response time, location of OHCA occurrence, and level of intervention by emergency medical technicians (EMTs) [4,5,6,7]. Additionally, some post-resuscitation care can impact the prognosis of OHCA, such as target temperature management (TTM), coronary angiography, extracorporeal membrane oxygenation (ECMO) intervention for ECMO facilitated resuscitation, and the level of hospital care [8,9,10,11,12,13,14]. However, patients who need TTM and ECMO-facilitated resuscitation are typically deeply sedated or comatose due to brain injury and therefore, difficult to evaluate post-resuscitation. Therefore, early prediction of OHCA prognosis remains crucial.

Previous studies have attempted to develop clinical decision rules to predict the outcome of OHCA, such as the OHCA score, TTM, and cardiac arrest hospital prognosis (CAHP) score [15,16,17]. Among available scores, the CAHP score seemed to present relatively better sensitivity and specificity to predict the neurologic outcome of OHCA [18,19]; however, the score can be relatively complex and clinical use is limited in emergency department (ED) settings. Recently, Pareek et al. have developed a practical risk score to predict the neurologic outcome of OHCA, termed MIRACLE2, which is substantially easier to apply [20]. The MIRACLE2 score consists of 7 items: missed witness OHCA, initial rhythm, reactivity of pupils, age, changing rhythm, low blood pH, and epinephrine administration. Compared to the CAHP score, MIRACLE2 exhibited a better receiver operating curve (ROC) along with a superior area under the curve (AUC). However, it should be noted that only OHCAs attributed to a primary cardiac cause were included in the MIRACLE2 study. Accordingly, we aimed to examine the clinical value of CAHP and MIRACLE2 scores for predicting neurological outcomes for OHCA. The primary outcome is to compare MIRACLE2 and CAHP scores in predicting neurological outcomes in OHCA.

## 2. Methods

### 2.1. Study Population

This retrospective observational study was conducted at a tertiary academic medical center with more than 2500 acute beds and an average of 72,000 adult ED visits annually. The medical records of adult patients (>17 years) were extracted from the ED administrative database from January 2015 to December 2019, using the International Classification of Diseases (ICD) Tenth Revision coding system (ICD code: I46). Two trained emergency physicians (EPs) reviewed electronic charts to identify patients with OHCA. OHCA due to trauma, burns, drowning, resuscitation not started due to pre-ordered “do not resuscitate” (DNR) orders, and patients who did not achieve return of spontaneous circulation (ROSC) after resuscitation were excluded.

### 2.2. Data Collection

Data were collected in two parts. The first part was collected from the EMS database, as described in our previous study [2]. Briefly, the EMS is a single-tiered system in Taiwan, and ambulance records were electronically stored in the EMS command center of each province. The EMS database included pre-hospital information, such as age, sex, witnessed OHCA, location of cardiac arrest (public or residential), initial rhythm, duration from cardiac arrest to CPR, and hospitals where patients were sent. The second part was collected from the ED administrative database of Kaohsiung Chang Gung Memorial Hospital. Data from the second part included disease underlying OHCAs, pupil reactivity, biochemistry data, epinephrine dose used, time of ROSC, and patient outcomes. A favorable neurologic outcome was defined as a cerebral performance category (CPC) of 1 to 2. CPC was measured at the time of hospital discharge. The present study was approved by our hospital’s institutional review board (number: 202100739B0) and was performed in accordance with the ethical standards of the 1964 Declaration of Helsinki and its later amendments.

### 2.3. Statistical Analysis

For independent variables, the results of continuous variables are presented as mean ± standard deviation (SD). The independent *t*-test and Mann–Whitney test were used to examine the difference in the distribution of continuous variables, and the chi-square test for independence was used to assess differences in the distribution of categorical variables between groups. ROC analysis with Youden Index was used to calculate the optimal cut-off value for CAHP and MIRACLE2 scores, which predict favorable neurologic outcomes. Logistic regression was used to analyze the statistically significant relationship between OHCA scores and outcomes. Univariate analysis was first performed, then a multivariate regression model was established to adjust for confounding factors. Odds ratios (ORs), 95% confidence intervals (CIs), and *p*-values were calculated using a logistic regression.

We assessed the classification performance of CAHP and MIRACLE2 scores using 2 × 2 contingency tables to generate estimates for sensitivity, specificity, positive predictive value (PPV), and negative predictive value (NPV). We calculated the 95% CIs for each proportion using the method described by Newcombe [16]. Statistical significance was set at *p* value < 0.05. All statistical analyses were performed using SPSS version 25.0 (IBM Corp., Armonk, NY, USA).

## 3. Results

The STROBE checklist for statistical reporting is presented in Appendix A. Figure 1 shows that 1043 adults (>17 years) were sent to our ED due to OHCA during the study period. We excluded patients who died due to burns, trauma, or drowning (n = 86); resuscitation not started due to a prescribing DNR order (n = 104); and patients who did not achieve ROSC (n = 643). Furthermore, 10 patients were excluded due to incomplete data, four cases due to missing “reactive pupil,” and six due to uncertain “collapse in BLS duration.”

After excluding subjects, 200 patients with OHCA were analyzed in the present study.

Demographic characteristics, pre-hospital factors, treatment in the ED, comorbidities, and calculated CAHP and MIRACLE2 scores are listed in Table 1. Overall, 33 patients with OHCA were discharged with favorable neurological outcomes. Patients who were discharged with favorable neurologic outcomes were associated with lower CAHP (*p* < 0.001) and MIRACLE2 scores (*p* < 0.001), lower epinephrine dose (*p* = 0.003), initial shockable rhythm (*p* = 0.011), received primary percutaneous cardiovascular interventions (PCI, *p* < 0.001), shorter time from collapse to basic life support (BLS), shorter time from BLS to ROSC (*p* < 0.001), and presented higher pH values (*p* = 0.005).

The results of the univariate analyses of prognostic factors are shown in Appendix A. Table 2 and Table 3 present the results of the multivariate logistic regression of OHCA adjusted for prognostic confounding factors, including CAHP and MIRACLE2 scores, epinephrine dose, primary PCI, and pH values. After adjusting for confounding factors, MIRACLE2 score (one additional point, OR = 0.654, 95% CI: 0.446–0.936, *p* = 0.020), epinephrine dose (OR = 0.894, 95% CI: 0.797–0.98, *p* = 0.015), and primary PCI (OR = 6.47, 95% CI: 2.584–16.769, *p* < 0.001) were significantly associated with favorable neurologic outcomes following OHCA. In addition, the CAHP score (one additional point, OR = 0.978, 95% CI: 0.963–0.992, *p* = 0.003), epinephrine dose (OR = 0.851, 95% CI: 0.714–0.981, *p* = 0.024), and primary PCI (OR = 8.479, 95% CI: 3.05–25.085, *p* < 0.001) were significantly associated with favorable neurologic outcomes. Goodness-of-fit using the Hosmer-Lemeshow test revealed no evidence of miscalibration (*p*-value for CAHP: 0.732; *p*-value for MIRACCLE2: 0.802).

Table 4 and Appendix A show the results of the ROC curve analysis and the calculated optimal cut-off values of MIRACLE2 and CAHP scores to predict favorable neurologic outcomes. The threshold of the MIRACLE2 score for favorable neurologic outcome was 5.5, with an AUC of 0.704 (0.610–0.797, *p* < 0.001), and the threshold of the CAHP score was 223.4, with an AUC of 0.773(0.688–0.858, *p* < 0.001).

Table 5 displays a 2 × 2 contingency table used to generate the following classification performance estimates for unfavorable neurological outcomes: Fifteen patients were excluded from the CAHP score classification because the time to ROSC due to ECMO intervention could not be recorded. When setting the cut-off value of the MIRACLE2 score to 5.5, we documented a sensitivity of 64.7% (95% CI, 56.9–71.9%), 66.7.0% specificity (95% CI, 48.2–82.0%), 90.8% PPV (95% CI, 85.6–94.2%), and 27.2% NPV (95% CI, 21.4–33.9%). On setting the CAHP score cut-off value as 223, a sensitivity of 68.2% (95% CI, 60.2–75.5%), 80.6% specificity (95% CI, 62.5–92.6%), 94.6% PPV (95% CI, 88.6–98.0%), and 33.8% NPV (95% CI, 23.2–45.7%) for unfavorable neurologic outcomes were achieved.

## 4. Discussion

In the present study, we compared a new score, termed MIRACLE2, with a frequently used score, CAHP, to predict neurologic outcomes in patients with OHCA. We found that CAHP was superior to MIRACLE2 in terms of AUC value, sensitivity, specificity, PPV, and NPV. Furthermore, we observed that the epinephrine dose employed during CPR, patients deemed suitable for PCI, MIRACLE2 score, and CAHP score were significantly associated with OHCA neurological outcomes.

Numerous scores have been designed to predict OHCA outcomes, such as OHCA, TTM, and CAHP scores [16,17,20,21]. A previous study has compared TTM, OHCA, and CAHP scores to predict the prognosis of OHCA. The authors reported that the AUC value of the TTM score was better than those of OHCA and CAHP scores [21]. Another study has compared OHCA, CAHP, Acute Physiology and Chronic Health Evaluation (APACHE) II, and Simplified Acute Physiology (SAPS) II scores to predict the prognosis of cardiac arrest [18]. Isenschmid et al. have reported the superior prognostic performance characteristics of the CAHP score than those of the OHCA, APACHE II, and SAPS II scores. Kim et al. have examined patients with OHCA who received TTM and compared the original CAHP, C-GRApH, and OHCA scores. The authors discovered that the CAHP score showed the best AUC when compared with OHCA and C-GRApH scores [22]. Notably, Pareek et al. have examined OHCAs attributed to suspected primary cardiac causes, treated at King’s College Hospital, and established a score to predict the outcome of OHCA, i.e., the MIRACLE2 score. Comparing MIRACLE2, TTM, OHCA, and CAHP scores, the authors revealed that the MIRACLE2 score displayed superior AUC, sensitivity, and PPV for predicting the neurologic outcome of OHCA [20]. However, in the present study, we found that both CAHP and MIRACLE2 scores were associated with the neurologic outcome of OHCA. We observed that the CAHP score afforded better discrimination ability in predicting neurologic outcomes than the MIRACLE2 score. In addition, the sensitivity, specificity, and PPV of the CAHP score were superior to those of the MIRACLE2 score. The discrepancy between our results and those reported by Pareek et al. could be attributed to differences in study groups examined. Notably, Pareek et al. included only OHCA patients with suspected primary cardiac causes. In the present study, we included OHCAs attributed to all underlying causes. Accordingly, 76.4% of patients underwent coronary angiography in the study reported by Pareek et al., whereas only 19% of patients with OHCA underwent coronary angiography in our cohort.

Although the discrimination ability of the CAHP score was better in our study, clinical validation was relatively limited. The duration of the “no-flow” interval (from collapse to CPR initiation) was occasionally difficult to estimate. In East Asian countries, the rates of witnessed OHCA were approximately 60% and 64% of cardiac arrest cases in Japan and Korea, respectively. The rates of witnessed OHCA were 81.1% and 91% in North America and France, respectively [23,24]. In our study, the rate of witnessed OHCA cases was 77.5%. Furthermore, the “low-flow” interval (from CPR to ROSC) was occasionally missed as the time of CPR initiation was omitted or ECMO was employed. Conversely, almost all items of MIRACLE2 can be obtained at hospitals, and we could easily calculate the MIRACLE2 score. Recently, a few studies have attempted to adjust the CAHP score to improve the ease of application. Wang et al. have excluded the duration of the “no-flow interval” from the CAHP score and observed that the AUC for neurological outcome was still acceptable [25]. However, large-scale studies validating the discrimination ability of such scores are lacking. Additional studies are required to establish scores with better clinical validation and discrimination.

Herein, we observed that a high epinephrine dose was related to a poor neurologic outcome. Previous studies have examined the relationship between the neurologic outcomes of patients with OHCA and pre-hospital epinephrine use, but the results were inconsistent. According to Nakahara et al., early epinephrine use during the pre-hospital phase, especially within 10 min of collapse, could be associated with favorable neurologic outcomes in bystander-witnessed patients with OHCA, regardless of cardiogenic or non-cardiogenic OHCA [26]. Ran et al. have reported a similar finding [27]. However, in a study by Loomba et al., pre-hospital epinephrine use was shown to be unrelated to favorable neurologic outcomes, with similar results reported in numerous other previous studies [28,29,30]. The differences between observed results might be explained by the duration of pre-hospital resuscitation time, and a longer pre-hospital CPR time also indicates greater epinephrine use. Kaji et al. have found that epinephrine use can be associated with favorable neurologic outcomes only when patients with OHCA receive dosages lower than 1.5 mg [31].

In the present study, patients with OHCA who underwent PCI exhibited favorable neurologic outcomes. Previous studies have reported similar results, although with slightly different patient selection settings. Bergman et al. included patients with OHCA presenting ROSC who were admitted to the hospital. The authors observed favorable neurologic outcomes in patients with OHCA presenting initial ventricular fibrillation (VF) rhythm, who then underwent PCI [32]. In Shavelle’s [33] study, although the survival rate of patients with OHCA was lower in the group with ST-elevation myocardial infarction (STEMI) than in the group without STEMI, the proportion of survival with favorable neurologic outcomes exceeded 50% (73%). Zimmermann et al. [34] have further highlighted that initiation of advanced cardiovascular life support (ACLS) within 6 min of collapse is a predictor of favorable neurologic outcomes in STEMI-complicated patients with OHCA. This result might be explained by the fact that VF is a common underlying cause of cardiogenic OHCA [35], and VF and cardiogenic OHCA are independent factors for favorable OHCA outcomes [36].

Some studies have focused on the impact of extracorporeal cardiopulmonary resuscitation (ECPR) for managing OHCA; however, the results remain inconclusive. Napp et al. selected 40 patients with OHCA presenting potentially favorable circumstances (witnessed collapse, initial shockable rhythm, without major comorbidities, bystander CPR, and age < 75 years) who received ECPR and analyzed the outcomes. The authors found that only three patients were discharged with favorable neurologic outcomes, and no significant differences were observed when compared with previous studies [37]. In contrast, Lunz et al. have examined 423 patients treated with ECPR at 5 different European intensive care units and found that 80 patients (19%) were discharged with favorable neurological outcomes [38]. In addition, Beyea et al. reviewed 75 studies and demonstrated that the ratios of favorable neurologic outcomes in the ECPR and conventional CPR were 8.3% to 41.6% and 1.5% to 9.1%, respectively [39]. However, a meta-analysis could not be performed owing to the significant heterogeneity in the study by Beyea et al. In the current study, we observed that ECMO did not afford a statistically significant difference. The case number was considerably small as only 15 patients received ECMO; further studies could help determine the impact of ECMO intervention.

The present study has some limitations. First, this was a retrospective observational study restricted to a single city. Second, the database was limited to a single-tiered EMS system, and the results may be different for other cities with different EMS systems. Third, some OHCA patients would be missed if they were brought to the hospital by family or healthcare facilities, not by EMS. Fourth, prehospital interventions, such as advanced airway management and mechanical devices for reanimation, were not recorded in this database, and these interventions might influence the outcome of OHCA. Furthermore, a history of PCI, arrhythmias, coronary artery bypass graft, and lactate values were not recorded, which might impact the prognosis of OHCA.

## 5. Conclusions

In the present study, we compared MIRACLE2 and CAHP scores to predict neurological outcomes for OHCA. We found that the CAHP score had better discrimination ability than the MIRACLE2 score. In addition, the CAHP score afforded a superior sensitivity, specificity, PPV, and NPV. We also found that epinephrine dose during CPR, patients suitable for PCI, and MIRACLE2 and CAHP scores were all significantly associated with OHCA neurological outcomes. However, the CAHP score was relatively difficult to apply despite the superior discrimination. Further investigations could help establish new scores with better discrimination and ease of application.

## Figures and Tables

**Figure 1 healthcare-10-00578-f001:**
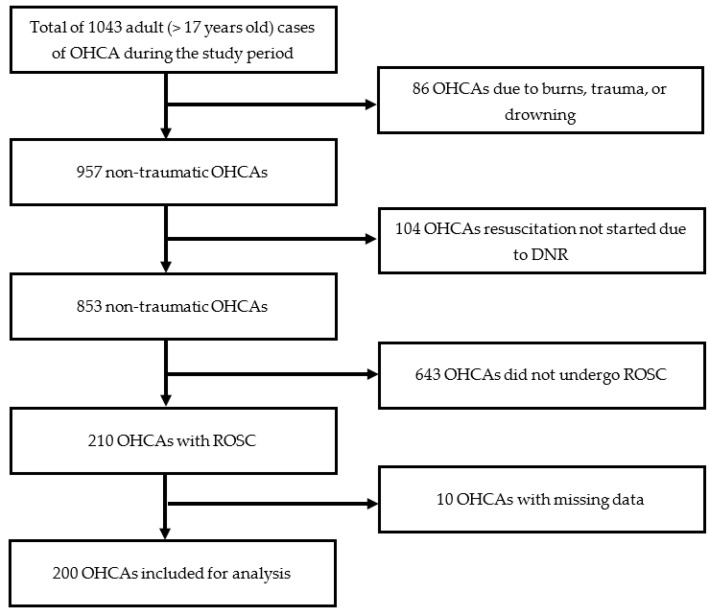
Flowchart of patients selection.

**Table 1 healthcare-10-00578-t001:** Demographic characteristics, pre-hospital factors, treatment in the ED, comorbidities, and calculated CAHP and MIRACLE2 scores of 200 OHCA patients with ROSC.

Demographic Characteristics	Poor Neurologic Outcome	Favorable Neurologic Outcome	*p*
	n = 167	n = 33	
Age	66.2 ± 16.2	64.6 ± 10.9	0.593
Male sex	101	19	0.756
MIRACLE2	5.9 ± 1.2	4.9 ± 1.4	<0.001
CAHP score	233.3 ± 36.1	194.8 ± 38.0	<0.001
Epinephrine dose (mg)	6.5 ± 6.0	4.1 ± 4.9	<0.001
Public location	44	12	0.242
Witness	133	22	0.103
Bystander CPR	60	10	0.536
Shockable rhyme	9	6	0.011
Primary PCI	23	15	<0.001
Collapse to BLS (min)	7.1 ± 5.0	5.3 ± 4.1	0.010
BLS to ROSC	27.4 ± 15.1	15.5 ± 10.6	<0.001
ECMO	13	2	0.731
pH	7.017 ± 0.179	7.115 ± 0.176	0.005
History of myocardial infarction	31	3	0.186
Diabetes	74	12	0.399
Malignancy	17	1	0.190
Liver cirrhosis	10	0	0.149
Renal insufficiency	56	11	0.982
COPD	18	2	0.409

Abbreviations: BLS, basic life support; CAHP, cardiac arrest hospital prognosis; COPD, chronic obstructive pulmonary disease; CPR, cardiopulmonary resuscitation; ECMO, extracorporeal membrane oxygenation; ED, emergency department; OHCA, out-of-hospital cardiac arrest; PCI, percutaneous cardiovascular interventions; ROSC, return of spontaneous circulation.

**Table 2 healthcare-10-00578-t002:** Adjusted odds ratios of MIRACLE2 for favorable neurologic outcome.

Variables	OR	95% CI		*p*
MIRACLE2	0.654	0.446	0.936	0.020
Epinephrine dose (mg)	0.894	0.797	0.98	0.015
Primary PCI	6.47	2.584	16.769	<0.001
pH	3.486	0.289	47.207	0.329

Abbreviations: CI, confidence interval; OR, odds ratio; PCI, percutaneous cardiovascular interventions.

**Table 3 healthcare-10-00578-t003:** Adjusted odds ratios of CAHP score for favorable neurologic outcome.

Variables	OR	95% CI		*p*
CAHP	0.978	0.963	0.992	0.003
Epinephrine dose (mg)	0.851	0.714	0.981	0.024
Primary PCI	8.479	3.05	25.085	<0.001
pH	0.966	0.059	17.109	0.981

Abbreviations: CI, confidence interval; CAHP, cardiac arrest hospital prognosis; OR, odds ratio; PCI, percutaneous cardiovascular interventions.

**Table 4 healthcare-10-00578-t004:** Receiver operating characteristic (ROC) curve analysis of the optimal threshold for predicting favorable neurologic outcome.

	Favorable Neurologic Outcomes				
Situation	Threshold (Points)	AUC	Lower	Upper	*p*
MIRACLE2	5.5	0.704	0.61	0.797	<0.001
CAHP score	223.4	0.773	0.688	0.858	<0.001

Abbreviations: AUC, area under the curve; CAHP, cardiac arrest hospital prognosis.

**Table 5 healthcare-10-00578-t005:** Sensitivity, specificity, positive predictive value, and negative predictive value of the risk scores for favorable neurologic outcome of OHCA.

MIRACLE_2_ Score (n = 200)			* CAHP Score (n = 185)		
Assessment Using MIRACLE_2_ > 5.5	Poor Neurologic Outcome	Favorable Neurologic Outcome	Assessment Using CAHP > 223	Poor Neurologic Outcome	Favorable Neurologic Outcome
No. of positive results	108	11	No. of positive results	105	6
No. of negative results	59	22	No. of negative results	49	25
Sensitivity, % (95% CI)	64.7 (56.9–71.9)		Sensitivity, % (95% CI)	68.2 (60.2–75.5)	
Specificity, % (95% CI)	66.7 (48.2–82.0)		Specificity, % (95% CI)	80.6 (62.5–92.6)	
PPV, %	90.8 (85.6–94.2)		PPV, %	94.6 (88.6–98.0)	
NPV, %	27.2 (21.4–33.9)		NPV, %	33.8 (23.2–45.7)	

Abbreviations: CI, confidence interval; CAHP, cardiac arrest hospital prognosis; NPV, negative predictive value; PPV, positive predictive value. * 15 patients were excluded from the CAHP score due to ECMO intervention.

## Data Availability

The datasets used and analyzed during the current study are available from the corresponding author on reasonable request.

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
