# Peer review of "Clinical Validation of Cardiac Arrest Hospital Prognosis (CAHP) Score and MIRACLE2 Score to Predict Neurologic Outcomes after Out-of-Hospital Cardiac Arrest"

_healthcare, 2022, doi:10.3390/healthcare10030578_

Round 1

Reviewer 1 Report

I read with great interest the manuscript by Wu et al. on the clinical validation of the CAHP Score and MIRACLE2 Score to predict neurologic outcomes after OHCA. However, I have some issues to be addressed by the authors:

Major issue:

The manuscript lacks completely of the discussion of the limitation of the study (i.e. the retrospective design of the study). I would suggest to add a paragraph at the end of the discussion and specify the limits in the conclusions and in the abstract.

Minor issues:

Line 66: When you specify your exclusion criteria, it is not clear whether you include trauma, burn and drowning patients when they are successfully resuscitated, please clarify.

Also, please explain the reasons for the choice to limit the study population.

Author Response

Major issue:

The manuscript lacks completely of the discussion of the limitation of the study (i.e. the retrospective design of the study). I would suggest to add a paragraph at the end of the discussion and specify the limits in the conclusions and in the abstract.

Response: Thank you for your valuable comment

  We added a part of limitation, in line 301-305 as following:

“There are some limitations in the present study. First, this was a retrospective observational study and restricted to a single city. Second, the database was limited to a single-tiered EMS system, and the results might be different for other cities with different EMS systems. Third, some OHCA patients would be missed if they were sent to hospital by family or healthcare facility, not by EMS.”

Minor issues:

Line 66: When you specify your exclusion criteria, it is not clear whether you include trauma, burn and drowning patients when they are successfully resuscitated, please clarify.

Also, please explain the reasons for the choice to limit the study population.

Response: Thank you for your valuable comment.

We edited the following sentences in line 69-73 as following:

  “OHCA due to trauma, burns, drowning, and resuscitation not started due to pre-ordered “do not resuscitate” (DNR) orders and patients who did not achieve re-turn of spontaneous circulation (ROSC) after resuscitation were excluded. We excluded these patients because the scores we validated also excluded these cases.” 

Reviewer 2 Report

The Jun Zuo Wu et al reported on clinical validation of CAHP score and MIRACLE2 score to predict neurologic outcomes after OHCA. Authors conclude that the CAHP score had better discrimination ability than the MIRACLE2 score, with a superior sensitivity, specificity, PPV, and NPV. Thez found that epinephrine dose during CPR, patients suitable for PCI, and MIRACLE2 and CAHP scores were all significantly associated with OHCA neurological outcomes.

The authors should be congratulated for a great work! I would suggest following adaptations of the work for publication in the Healthcare:

Abstract:

  1. Line 2: Introduce MIRACLE2 shortcut, in text also.
  2. Line 12-13: Please start the sentence with ‘The aim of the study’ or similar.
  3. Line 15: Please revise, what does ‘sent by emergency service’ mean?
  4. Please report on AUC and range with maximum of 2 decimals, the p value should be with 3 decimals.
  5. Please revise the lines from 20-27 to avoid double repeating of numbers 5.5 and 223.
  6. Keywords are missing

Introduction:

  1. Line 35: Please reconsider revision of this sentence. Even if the OHCA medical challenge is, it is not really emergency physician challenge due to poor prognosis but the intensive care unit afterwards. Maybe you could revise similar to ’Out-of-hospital cardiac arrest (OHCA) remains challenging, given the poor prognosis of OHCA.’ Or similar.
  2. Line 36: Please avoid using of word ‘Reportedly‘
  3. Line 38-40: Please arrange factors by importance (bystander reanimation has much more effect compared to the EMS response time)
  4. Line 45-46: These two sentences are not connected and state completely different opinions. Furthermore, outcome of OHCA is most often evaluated after hospital discharge or number of patients discharged alive. Use of ECMO or TTM can only support all intensive care measurements. However, this is not important for prognostication of OHCA related to the level of consciousness after resuscitation. It is rather that patients needing ECMO support have more serious underlying disease.

Methods:

  1. Please revise your statistical methods reporting in more details, reporting which test was used for which type of variables (Depending on the normality of the distribution and the type of variables)
  2. Please revise: Statistical significance was set at p < 0.05. -> ‘was set at a p values of less than 0.05’, or ‘a significance level of 0.05 was applied’
  3. Line 79-80: at which point was the CPC measured – please report.

Results:

  1. Please provide figure of flow chart for patients selection.
  2. Please report on missing data for every variable. Also for multivariate model. And how did you deal with the missing data?
  3. Please consider using one of the checklists for statistical reporting (e.x. STROBE)
  4. Please revise line 100: ‘We excluded deaths due to burns’, you excluded patients who died due to …
  5. Please revise the number of patients and exclusion reasons, as the result is 210, not 200.
  6. Table 1: Public location is reported 2 times. Please revise
  7. Table 1: please provide pH with 3 decimals.
  8. Table 1: Comorbidities as cardiovascular or stroke are missing? History of myocardial infarction, arrhythmias, PTCA or CABG.
  9. Table 1: Epinephrine dose – where the data normal? How did u select statistical test? Would man-Whitney be more appropriate?
  10. Table 1: Do you have data on low-flow/no-flow? Data on delay time (from arrest to CPR start)?
  11. Table 1: Data on time to ROSC?
  12. Table 1: data on number of witnessed OHCA?
  13. Table 1: data on initial rhythm?
  14. Table 1: Data on ROSC out of hospital, transport under reanimation, use of mechanical devices for reanimation?
  15. Table 1: Number of patients with ROSC with EMS outside of hospital?
  16. Table 1: Intubation?
  17. Do you have data on initial lactate and pH?
  18. Do you have data on Initial Temperature? Pupillary / corneal reflex?
  19. Please provide results in the table from lines 78-79 in methods.
  20. Number of patients with acute STEMI?
  21. Please provide table with the underlying disease most probably leading to the cardiac arrest and outcome.
  22. Line 118: spelling/grammar
  23. Did you perform univariate analyses of factors before including them in the multivariate model? Please provide the table as supplementary.
  24. How did you select the variables for multivariable model? Please describe in methods.
  25. Line 121: always report p value with 3 decimals. Revise the whole paper.
  26. Please provide a clear table presenting patients with the bad and good neurological outcome, with scores from MIRACLE2 (all calculated subsections if available, for example 1. missed number of patients with score 1 and number of patients with score 0; Initial rhythm number of patients with the score 1 and number with the score 0 etc.). And the same for CHAP score.
  27. Table 4: Please provide both ROC figures as supplementary.
  28. Lines 146-154: Please revise reporting as it should be in the results section, check for English syntax and grammar.
  29. Table 5: please perform technical revision of the table
  30. 15 patients needed ECMO in your study, provide this finding in the results section tables. What was the mortality for ECMO patients?

Discussion:

  1. Line 194-195 is not clear, please revise. What is the rate of witnessed OHCAs? What is the rate in your study?

Round 2

Reviewer 1 Report

I would like to thank the authors for their response, which completely clarified my concerns.

I still have one minor issue regarding the manuscript introduction:

Line 42-43: "Additionally, some post-resuscitation care can impact the prognosis of OHCA, such as target temperature management (TTM), coronary angiography, extracorporeal membrane oxygenation (ECMO) intervention for ECMO facilitated resuscitations"

I think that more bibliography could be provided to support this sentence.

I would suggest 3 recent meta-analyses for the TTM:

  • Fernando et al., Intensive Care Med. (2021), https://doi.org/10.1007/s00134-021-06505-z
  • Sanfilippo et al, J. Clin. Med. (2021), https://doi.org/10.3390/jcm10173943
  • A. Granfeldt et al., Resuscitation (2021), https://doi.org/10.1016/j.resuscitation.2021.08.040

Coronary angiography:

  • Verma et al., JACC Cardiovasc Interv. (2020), 10.1016/j.jcin.2020.07.018
  • Camuglia et al., Resuscitation. (2014), 10.1016/j.resuscitation.2014.08.025. Epub 2014 Sep 4 

ECPR:

  • Halenarova et al., Resuscitation. (2022), 10.1016/j.resuscitation.2021.11.015
  • Lunz et al., Intensive Care Med. (2020), 10.1007/s00134-020-05926-6

Reviewer 2 Report

The quality of this work improved significantly, and authors should be congratulated! However, the following points should be still checked. Furthermore, authors are kindly advised to take some time and read the manuscript carefully to improve syntax and grammar, this is crucial for publication of scientific work. Finally, mistakes due to fast revision may be not seen as a positive (supplementary figure 1, spelling mistakes in revised text, etc).

Line 45-46 "under more severe unconscious" - this is wrong phrase, please revise. These patients are deeply sedated or due to brain injury deeply comatose and therefore GCS 3.

Line 72: "We excluded these patients because the scores we validated also excluded these cases" This saying is not needed.

Line 85: Primary outcome can be stated at the end of introduction or at the Methods section, as the part of introduction to methods.

Line 109: Revise writing/grammar.

Line 110: Revise, it is not clear what are you saying here.

Line 110: Patients did not visit your ED, they were admitted, brought, or delivered to the ED during CPR. Visit consider something else. Also, Data on ROSC out of the hospital, transport under reanimation, use of mechanical devices for reanimation, data on the airway, have data on Initial Temperature, Pupillary/corneal reflex. 

Figure 1: Shortcuts are missing

Line 158: grammar, please revise.

Table 1: Epinephrine dose – where is the data normal? How did u select a statistical test? Would man-Whitney be more appropriate?

  Response: Thank you for noticing this part. We originally used the independent t-test to calculate, the resulting p-value was 0.03. Now, we calculated the p-value using the Mann-Whitney U test, and the resulting p<0.001. We revised the value in table 1 and revised the statistical analysis in the method part.

---> did you check the normality of data or you were looking only at which p-value fits better your results? How did you decide which test to use?

Line 304: Patients are not sent to the hospital.

Please provide as limitation in the limitations section missing of lactate, history of arrhythmias, PTCA, or CABG.

STROBE statement: What does "V" mean? The page number should be reported.

Provide the univariate analyses of Witness, Public locations, etc from table 1.

The authors should be congratulated for the new figure, however the figure is missing a complete title. Supplementary figure 1: Please finish the description of the figure...
